# Illustrating the importance of edge constraints in backbones of bipartite projections

Zachary P. Neal [ID]*, Jennifer Watling Neal[ID]

Psychology Department, Michigan State University, United States of America

* zpneal@msu.edu

**Data Availability Statement:** The data and code necessary to reproduce the results reported below are available at https://osf.io/83bxq/.

**Funding:** The author(s) received no specific funding for this work;.

## Abstract

Bipartite projections (e.g., event co-attendance) are often used to measure unipartite networks of interest (e.g., social interaction). Backbone extraction models can be useful for reducing the noise inherent in bipartite projections. However, these models typically assume that the bipartite edges (e.g., who attended which event) are unconstrained, which may not be true in practice (e.g., a person cannot attend an event held prior to their birth). We illustrate the importance of correctly modeling such edge constraints when extracting backbones, using both synthetic data that varies the number and type of constraints, and empirical data on children's play groups. We find that failing to impose relevant constraints when the data contain constrained edges can result in the extraction of an inaccurate backbone. Therefore, we recommend that when bipartite data contain constrained edges, backbones be extracted using a model such as the Stochastic Degree Sequence Model with Edge Constraints (SDSM-EC).

## Introduction

Unipartite networks are often measured using bipartite projections, where the edges capture nodes' shared characteristics. For example, a unipartite social network can be measured using a bipartite projection in which edges capture individuals' numbers of event co-attendance [1]. Because the edge weights in a bipartite projection provide noisy information about whether two individuals likely have a relationship, backbone extraction models are recommended to filter these noisy edges [2]. Most backbone extraction models impose the assumption that it is logically possible for any edge in the bipartite network to be either present or absent. For example, in a bipartite person-event network from which a social network can be obtained via projection, these models assume that any person could have attended any event, and any person could have refrained from attending any event. However, this assumption may be violated in some cases. For example, a person cannot have attended an event that occurred before they were born; such an edge is logically impossible.

A variant on an existing backbone model, the Stochastic Degree Sequence Model with Edge Constraints (SDSM-EC) [3], was recently developed that can relax this assumption by imposing constraints on certain bipartite edges that are known to be logically necessary or logically

**Competing interests:** The authors have declared that no competing interests exist.

impossible. However, it remains unknown to what extent imposing such constraints on a backbone model impacts the resulting backbone, and thus whether using such a model is necessary. Therefore, in this study, we investigate the importance of imposing edge constraints on backbone extraction models, using both synthetic and empirical data. We use synthetic data to systematically vary the number of type of constrained edges, and use empirical data on observed children's play groups that are known to contain three types of constrained edges: some edges are logically necessary due to the observational data collection method, some edges are logically impossible due to age-segregated classrooms, and some edges are logically impossible due to children's different attendance times. For both types of data, we compare a backbone extracted using the SDSM-EC, and therefore which yields a more accurate backbone by accounting for all these edges constraints, to backbones extracted using a model that accounts for none or only some of these constraints. We find that backbones which fail to account for constrained edges can be quite different from those that do, and therefore can lead to inaccurate backbones. We also find that the degree of inaccuracy is directly associated with the number of edge constraints that are imposed. These findings suggest that, when extracting the backbone of a projection of a bipartite network that contains constrained edges, a backbone extraction model such as SDSM-EC should be used.

The paper is organized in four sections. In the background section, we review bipartite backbone models, describe the types of constrained edges that can occur in bipartite data, and discuss the importance of backbone models considering such edges. In the synthetic illustration section, we describe a set of numerical experiments that investigate the importance of edge constraints in backbones, varying the number and type of constraints that are present. In the empirical illustration section, we compare backbones that were extracted taking into account all, some or none of the edge constraints present in observational data on children's interactions. In the discussion section, we summarize the findings and offer recommendations for researchers wishing to extract the backbone from bipartite projections.

## Background

Unipartite networks can be difficult to measure directly. In such cases, researchers often measure them indirectly using the projection of a bipartite network. Given a bipartite network $\mathbf{B}$, where $B_{ik} = 1$ if type-A node $i$ is connected to type-B node $k$, a bipartite projection $\mathbf{P}$ can be constructed via projection as $\mathbf{P} = \mathbf{B}\,\mathbf{B}'$, where $\mathbf{B}'$ denotes the transpose of $\mathbf{B}$, and where $P_{ij}$ is the number of type-B nodes $k$ shared by type-A nodes $i$ and $j$. For example, given a bipartite network describing which people (type-A nodes) attended which events (type-B nodes), a bipartite projection describes which people attended the same events as others, where the edge weights indicate the number of co-attended events [1, 4].

In some cases, a bipartite projection can be analyzed as a unipartite network, albeit one that was measured indirectly. However, the interpretation of a bipartite projection is challenging because the edge weights $P_{ij}$ provide noisy signals about whether nodes $i$ and $j$ are likely to have a relationship [2]. Two examples serve to illustrate the potential noisiness of projection edge weights. Consider two people who attended the same five small dinner parties, and no other events; their edge weight of 5 may be small, but nonetheless offers strong evidence they have a relationship. In contrast, consider two people who have each attended 100 large concerts, among which they happened to overlap at 10 concerts; their edge weight of 10 may be larger, but provides little evidence they have a relationship. Therefore, when seeking to interpret a bipartite projection as an indirectly-measured unipartite network, it is often preferable to focus on the 'backbone' of a bipartite projection, which is an unweighted subgraph of the projection that contains only those edges whose weights are statistically significantly large.

Many models exist for determining which edges in a projection have statistically significant weights, and therefore for extracting the backbone. However, they all rely on a common process. The weight of a given edge in the projection $P_{ij}$ is compared to the distribution of weights of a corresponding edge $P_{ij}^*$ obtained via projection of a random bipartite network $\mathbf{B}^*$. Using a conventional Null Hypothesis Significance Test (NHST), if $P_{ij}$ is in the upper tail of the distribution of $P_{ij}^*$, then it is retained in the backbone. Conceptually, an edge is retained in a backbone if its weight is significantly larger than would be expected in a projection obtained from a random bipartite network.

Backbone models differ only in their null models, and specifically how they construct and constrain the random bipartite network $\mathbf{B}^*$. The mathematical specifications and algorithms for these models are presented in detail elsewhere [5, 6]; we refer readers to these materials for technical details. In this work, we focus on the stochastic degree sequence model (SDSM) for four reasons: it is widely-used [7–11], fast [2], accurate [5, 12], and is the basis for the SDSM-EC variant that can impose edge constraints [3]. In the SDSM, $\mathbf{B}^*$ is constrained to ensure that its *expected* degree sequences match those of $\mathbf{B}$. For example, if person $i$ was observed to attend 10 events and event $k$ was observed to have 25 attendees in $\mathbf{B}$, then *on average* person $i$'s degree equals 10 and event $k$'s degree equals 25 across all possible $\mathbf{B}^*$.

When a random bipartite network $\mathbf{B}^*$ is generated using SDSM, the degree sequences of $\mathbf{B}$ are constrained as described above, but the positions of the 0s (missing edges) and 1s (present edges) are otherwise random. This implicitly imposes the assumption that any given type-A node may or may not be connected to any given type-B node. However, this assumption may be violated in practice. Table 1 provides a simple example of a bipartite network describing three network researchers (type-A nodes) and their attendance at three different events (type-B nodes). Most of the edges are unconstrained and therefore are consistent with SDSM's assumption: it is logically possible that the person attended the event, and it is also logically possible that the person did not attend the event. For example, Frank Harary *could have* attended the 1985 Academy Awards, no matter how improbable that might have been.

However, Table 1 also illustrates two different types of constrained edges: prohibited edges and required edges. Prohibited edges are edges that are logically impossible; they will be missing in the observed $\mathbf{B}$, but they must also be missing in any $\mathbf{B}^*$. In this example, the edge between Duncan Watts and the 1965 conference is prohibited because Duncan Watts was born in 1971, and therefore it is logically impossible for him to have attended this event. Conversely, required edges are edges that are logically necessary; they will be present in the observed $\mathbf{B}$, but they must also be present in any $\mathbf{B}^*$. In this example, the edge between Paul Erdős and the 1995 keynote is required because Paul Erdős was the speaker at this event, and therefore it is logically necessary for him to have attended.

To correctly model the observed data, a backbone model must impose these constraints on $\mathbf{B}^*$. The SDSM-EC is identical to SDSM in every respect, except that it can also constrain $\mathbf{B}^*$ to exclude prohibited edges and include required edges. The mathematical specification of SDSM-EC is presented in detail elsewhere [3] and is implemented in the R `backbone` package [13]; we refer readers to these materials in the *Supplementary Information* available at

**Table 1. Example bipartite network with constrained edges.**

|  | A complex network conference in 1965 | The Academy Awards in 1985 | A keynote by Erdős in 1995 |
| --- | --- | --- | --- |
| **Paul Erdős** | Unconstrained | Unconstrained | Required (gave the lecture) |
| **Frank Harary** | Unconstrained | Unconstrained | Unconstrained |
| **Duncan Watts** | Prohibited (not born yet) | Unconstrained | Unconstrained |

https://osf.io/83bxq/ for technical details. Although SDSM-EC makes it possible to impose constraints on specific edges in $\mathbf{B}^*$, it remains unknown whether doing so yields a meaningfully different backbone than simply ignoring any constrained edges that may exist in the data. Therefore, in this study, we use the SDSM-EC to investigate the following research question: *To what extent do backbones extracted using a model that imposes edge constraints differ from backbones extracted using a model that ignores constrained edges*? Or, more practically, we ask: *How important is it for researchers to impose relevant edge constraints when extracting a backbone from a bipartite projection*?

## Synthetic illustration

### Methods

To illustrate the importance of edge constraints in the backbones of bipartite projections, we conduct a series of numerical experiments that vary the percentage of edges that are either required or prohibited. First, we generate a random bipartite network containing 20 type-A nodes and 100 type-B nodes [14, 15]. Second, we treat a randomly chosen X% of present edges as required, and a randomly chosen Y% of absent edges as prohibited, where we independently vary X and Y between 0% and 90%. Third, we extract the backbone of the projection of these data using the conventional SDSM (which ignores these constraints) and the SDSM-EC (which models these constraints), using $\alpha = 0.05$ in each case. Fourth, we compare the two backbones by computing the correlation coefficient, Cohen's kappa, the Jaccard index [16]. Finally, for each combination of X and Y, we repeat the experiment 100 times, and in the results focus on the average similarity between the constrained and unconstrained backbones for each experimental condition. In the results reported below, we focus on the correlation coefficient to measure similarity; the other similarity indices are provided in the *Supplementary Information* available at https://osf.io/83bxq/ and yield similar results.

### Results

Fig 1 displays the correlation between a backbone extracted using SDSM that ignores edge constraints, and a backbone extracted using SDSM-EC that considers edge constraints, varying the percentage of edges that are constrained as required or prohibited. These results indicate that when bipartite data contain relatively few constrained edges, SDSM and SDSM-EC yield similar backbones. For example, when only 10% of edges are constrained because they are required, and only 10% of edges are constrained because they are prohibited, SDSM and SDSM-EC extracted backbones are quite similar ($r = 0.93$). In contrast, when bipartite data contain many constrained edges, SDSM and SDSM-EC yield very different backbones. For example, when 90% of edges are constrained because they are required, and 90% of edges are constrained because they are prohibited, SDSM and SDSM-EC extracted backbones are quite different ($r = 0.27$). Collectively, these results highlight that when bipartite data contain constrained edges, it is important to correctly model these constraints when extracting the backbone by using SDSM-EC because failing to do so (i.e., failing to correctly model the data) yields a backbone that is quite different.

## Empirical illustration

### Methods

To investigate the importance of edge constraints in the backbones of bipartite projections, we use observational data on children's participation in play groups [17–19]. The researchers who originally collected these data used a random scan technique to record play groups among 53

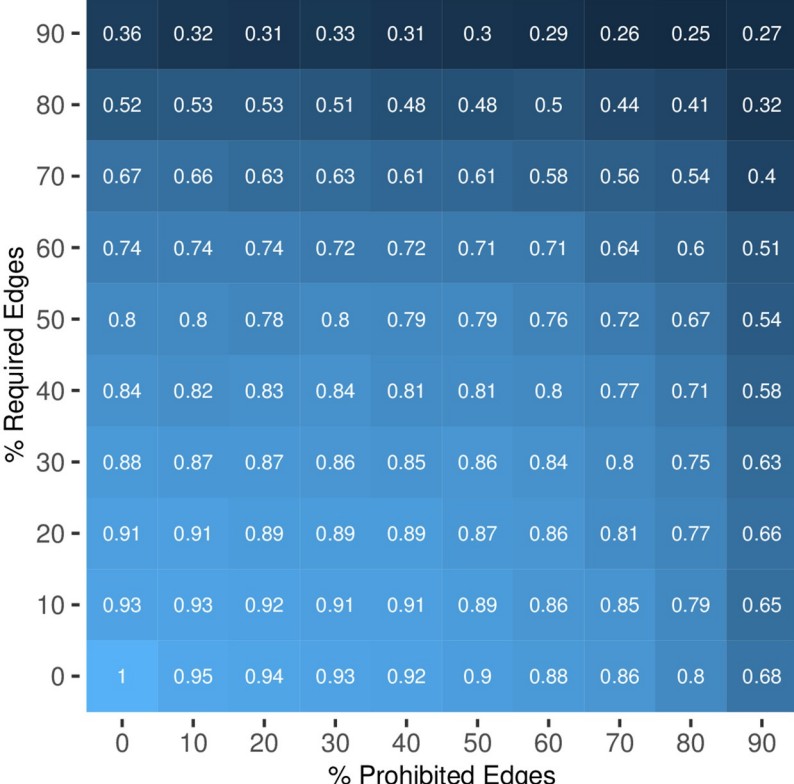

**Fig 1. Similarity (correlation coefficient) between a backbone extracted using SDSM that ignores edge constraints, and a backbone extracted using SDSM-EC that considers edge constraints.**

children in a preschool in the Midwestern United States between January and April 2013. During an observation period, trained observers rotated through a randomly ordered list of children in the classroom. Each randomly-selected *focal* child was observed for a 10 second 'scan'. If the focus child was involved in social play (i.e., interactive play with one or more other children) during this 'scan', then the other members of the play group were recorded. A total of 1829 play groups were observed. The collection of these data, and a waiver of parental consent and student assent, was approved by the Michigan State University Institutional Review Board (11-1198). For this analysis, we organize these data as a 53 × 1829 biadjacency matrix **B** where $B_{ik} = 1$ if child $i$ was observed in play group $k$.

From these data, it is possible to construct the adjacency matrix of a unipartite, weighted social network as **P** = **B B′**, where $P_{ij}$ indicates the number of times that child $i$ and child $j$ were observed in the same play group. Because such a network is noisy and dense, backbone extraction can be employed to obtain a sparser social network that retains edges between pairs of children who were observed playing together more often than expected at random. However, backbone extraction is complicated because characteristics of this data collection procedure and this setting mean that the bipartite data contains three distinct types of constrained edges. Although the observations were collected during periods of unstructured free-play, which meant that children could mostly play in any groups they chose, some child-group pairings were logically necessary or logically impossible.

First, the scan-based method used to collect these observations meant that the randomly-selected focal child must be a member of a play group observed during a given scan. Therefore,

**Table 2. Characteristics of observed bipartite data with varying edge constraints.**

| Nodes Observed | | Constraints Imposed | | | Missing Edges | | Present Edges | |
|---|---|---|---|---|---|---|---|---|
| Children | Groups | Focal | Age | Time | Unconstrained | Prohibited | Unconstrained | Required |
| 53 | 1829 | | | | 92420 | 0 | 4517 | 0 |
| 53 | 1829 | X | | | 92420 | 0 | 2688 | 1829 |
| 53 | 1829 | | X | | 44409 | 48011 | 4517 | 0 |
| 53 | 1829 | | | X | 80648 | 11776 | 4513 | 0 |
| 53 | 1829 | X | X | | 44409 | 48011 | 2688 | 1829 |
| 53 | 1829 | X | | X | 80648 | 11776 | 2684 | 1829 |
| 53 | 1829 | | X | X | 38477 | 53947 | 4513 | 0 |
| 53 | 1829 | X | X | X | 38477 | 53947 | 2684 | 1829 |

the edge between a focal child and the associated play group is a *required* edge. Taking account of this type of constrained edge in a backbone extraction model requires that $\forall\ \mathbf{B}^*, B^*_{focal,\ k} = 1$.

Second, the preschool was organized into two age-segregated classrooms that met in separate spaces and never interacted. This segregation meant that children in the three-year-old classroom could not play with children in the four-year-old classroom. Therefore, the edge between a child in one classroom and a play group being observed in the other classroom is a *prohibited* edge. Taking account of this type of constrained edge in a backbone extraction model requires that $\forall\ \mathbf{B}^*, B^*_{3yo\ child,\ 4yo\ group} = 0$ and $B^*_{4yo\ child,\ 3yo\ group} = 0$.

Third, children could attend the preschool in the morning only (AM), in the afternoon only (PM), or for the full day (FULL). This attendance schedule meant that a group observed in the morning could include AM and FULL children, but not PM children. Likewise, a group observed in the afternoon could include PM and FULL children, but not AM children. Therefore, the edge between an AM child and a play group being observed in the afternoon, or the edge between a PM child and play group being observed in the morning, is a *prohibited* edge. Taking account of this type of constrained edge in a backbone extraction model requires that $\forall\ \mathbf{B}^*, B^*_{AM\ child,\ PM\ group} = 0$ and $B^*_{PM\ child,\ AM\ group} = 0$.

Table 2 describes eight different bipartite networks that can be defined by including different combinations of constrained edges. All bipartite networks contain 53 nodes representing children and 1829 nodes representing play groups. Additionally, all bipartite networks contain a total of 4517 present edges (i.e., a given child was observed playing in a given group) and a total of 92,420 missing edges (i.e., a given child was *not* observed playing in a given group). However, they differ in their number of constrained edges. The bottom row indicates that when constraints are imposed that account for required edges associated with focal children, and that account for the prohibited edges associated with age and time, the bipartite network contains 53,947 constrained missing edges (i.e., prohibited, 58% of all missing edges) and 1829 constrained present edges (i.e., required, 40% of all present edges). When fewer constraints are imposed, the bipartite network contains correspondingly fewer constrained edges.

To explore the importance of edge constraints, we use SDSM-EC to extract the backbone from the projection of each bipartite network shown in Table 2. In each case, we use $\alpha = 0.13$, following prior empirical work analyzing these data [3]. Because the SDSM-EC takes account of edge constraints known to be present in the data, the backbone extracted from the maximally-constrained bipartite network (bottom row) most correctly models the data. Therefore, we compare each extracted backbone to this correctly-modeled backbone using three common indices of binary classification accuracy: the correlation coefficient, Cohen's kappa, the Jaccard index [16]. In addition, we report the values in the confusion matrix for each backbone

comparison, which consists of the number of true positives (edge is present in both backbones), true negatives (edge is missing in both backbones), false positives (edge is present in less-constrained backbone, but missing in the maximally-constrained backbone), and false negatives (edge is missing in less-constrained backbone, but present in the maximally-constrained backbone). The data and code necessary to reproduce the results reported below are available at https://osf.io/83bxq/.

## Results

Fig 2 illustrates the backbones extracted from bipartite data using models that impose different types of edge constraints. Panel H is the backbone obtained by imposing all relevant edge constraints, correctly modeling these data, and therefore serves as the backbone against which we compare other backbones that incorrectly model these data. The other panels illustrate that when fewer constraints are imposed, the resulting backbone can be quite different, often including many erroneous edges, and sometimes missing some true edges.

Table 3 summarizes the comparisons of the maximally-constrained backbone to other backbones extracted by imposing fewer constraints. We observe that the maximally-constrained backbone and the fully-unconstrained backbone have a correlation of only $\rho = 0.752$. Additionally, the fully-unconstrained backbone contains 62 false positives (FP), which corresponds to a 40.5% false discovery rate (FDR), which is unacceptably high. This highlights that failing to account for any constrained edges when they exist in the data results in extracting an inaccurate backbone. When accuracy is measured using correlation, the accuracy of a backbone is almost perfectly correlated with the number of relevant edge constraints that are imposed by the backbone extraction model ($\rho = 0.979$). This suggests that when the bipartite data contain constrained edges, the accuracy of the extracted backbone increases as more edge constraints are imposed by the backbone extraction model. Although we focus on correlation as an index of accuracy, the same patterns hold for other indices of accuracy.

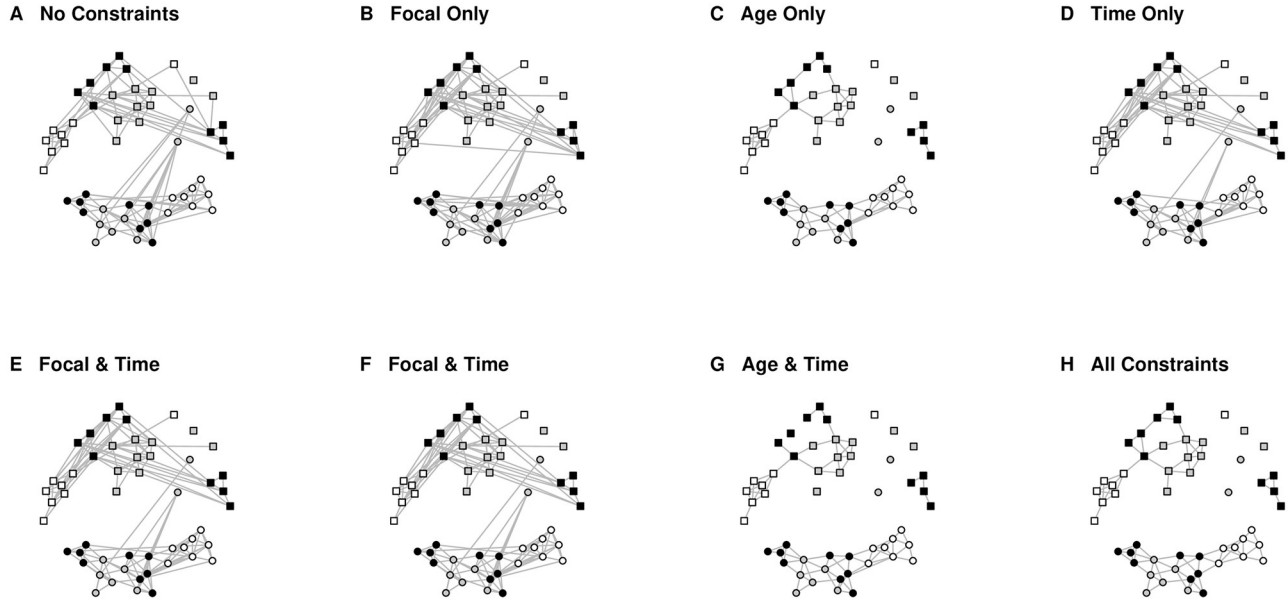

**Fig 2. Backbones extracted from bipartite data with varying edge constraints.** Attendance time is represented by node color: black = full day attendance, gray = AM only attendance, white = PM only attendance. Age is represented by node shape: circles = 3-year-olds, squares = 4-year-olds.

**Table 3. Comparing backbones extracted from bipartite data with varying edge constraints.**

| Constraints Imposed | | | Similarity to All-Constraints Backbone | | | Confusion Matrix | | | |
| --- | --- | --- | --- | --- | --- | --- | --- | --- | --- |
| Focal | Age | Time | Correlation | Kappa | Jaccard | TP | TN | FP | FN |
| | | | 0.752 | 0.723 | 0.595 | 91 | 1225 | 62 | 0 |
| X | | | 0.699 | 0.657 | 0.523 | 91 | 1204 | 83 | 0 |
| | X | | 0.954 | 0.954 | 0.918 | 89 | 1281 | 6 | 2 |
| | | X | 0.781 | 0.758 | 0.636 | 91 | 1235 | 52 | 0 |
| X | X | | 0.946 | 0.944 | 0.901 | 91 | 1277 | 10 | 0 |
| X | | X | 0.755 | 0.726 | 0.599 | 91 | 1226 | 61 | 0 |
| | X | X | 0.952 | 0.951 | 0.913 | 84 | 1286 | 1 | 7 |
| X | X | X | – | – | – | – | – | – | – |

TP = True positives, TN = True negatives, FP = False positives, FN = False Negatives

## Discussion

Bipartite projections (e.g., event co-attendance) are often used to measure unipartite networks of interest (e.g., social interaction). Backbone extraction models can be useful for reducing the noise inherent in bipartite projections. However, they typically assume that the bipartite edges (e.g., who attended which event) are unconstrained, which may not be true in practice. In some cases, bipartite edges may be either required (e.g., the host of an event must attend it) or prohibited (e.g., a person cannot attend an event prior to their birth).

In this study, we used both synthetic and empirical data to investigate the importance of using a backbone extraction model capable of imposing edge constraints when the bipartite data contain such constrained edges. The synthetic data illustrated that as more constrained edges exist, the backbones extracted using SDSM (which ignores these constraints) and SDSM-EC (which models these constraints) become less similar. In the empirical data, when none of these constraints were imposed by the backbone extraction model, the resulting backbone exhibited poor accuracy ($\rho = 0.752$) and a high false discovery rate ($FDR = 40.5\%$). As additional constraints were imposed, the accuracy improved as a nearly-perfect linear function of the total number of constrained edges ($\rho = 0.979$). These results highlight that when constrained edges exist in observed bipartite data, it is important to impose edge constraints on the backbone extraction model.

These results must be viewed in light of some limitations, which point to directions for future research. First, in our empirical illustration we treat the maximally-constrained backbone as a pseudo-ground truth because it incorporates all types of edge constraints that exist in these bipartite data, and therefore correctly models the data. However, it is not a true ground truth because it does not represent an independently and directly empirically measured unipartite network. If it had been possible for the original researchers to directly measure the social network among these children, they would have simply done so, rather than undertake the laborious process of observing play groups [17]. Nonetheless, these findings should be replicated in a context where (a) a unipartite network of interest is directly measured that can serve as a ground-truth, (b) an associated bipartite network from which a backbone can be extracted is independently measured, and (c) the location of multiple types of constrained edges are known in the bipartite data. Second, we examined the importance of edge constraints when extracting the backbone of a bipartite projection using the SDSM. Other backbone models exist, most notably the fixed degree sequence model (FDSM), which potentially offers greater statistical power at the expense of computational complexity [5]. There is

currently no way to incorporate edge constraints into the FDSM null model, however future research may explore ways to modify the curveball [20] or fastball [21] randomization algorithms to do so.

These limitations notwithstanding, our results offer some preliminary guidance for researchers seeking to measure a unipartite network using the backbone of a bipartite projection. First, carefully consider whether the data collection methods or contextual features of the network imply that some edges in the bipartite network are constrained by being either prohibited or required. Second, if constrained edges exist, use SDSM-EC to extract the backbone, imposing constraints on as many of the constrained edges as possible. Third, if constrained edges exist but it is not possible to impose constraints on all of them because the locations of certain constrained edges is unknown, interpret the resulting backbone with caution because it will be inaccurate, where the magnitude of inaccuracy is a function of the number of constrained edges that are not properly constrained in the null model.

## Acknowledgments

We thank Emily Durbin for her assistance collecting the empirical data.

## Author Contributions

**Conceptualization:** Zachary P. Neal, Jennifer Watling Neal.

**Data curation:** Jennifer Watling Neal.

**Formal analysis:** Zachary P. Neal.

**Investigation:** Zachary P. Neal, Jennifer Watling Neal.

**Software:** Zachary P. Neal.

**Visualization:** Zachary P. Neal.

**Writing – original draft:** Zachary P. Neal.

**Writing – review & editing:** Zachary P. Neal, Jennifer Watling Neal.

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
