## [Decision Letter · Decision Letter 0]

5 Apr 2024

PONE-D-24-08069The importance of edge constraints in backbones of bipartite projectionsPLOS ONE

Dear Dr. Neal,

Thank you for submitting your manuscript to PLOS ONE. After careful consideration, we feel that it has merit but does not fully meet PLOS ONE’s publication criteria as it currently stands. Therefore, we invite you to submit a revised version of the manuscript that addresses the points raised during the review process. Please submit your revised manuscript by May 20 2024 11:59PM. If you will need more time than this to complete your revisions, please reply to this message or contact the journal office at plosone@plos.org. Please include the following items when submitting your revised manuscript:A rebuttal letter that responds to each point raised by the academic editor and reviewer(s). You should upload this letter as a separate file labeled 'Response to Reviewers'.A marked-up copy of your manuscript that highlights changes made to the original version. You should upload this as a separate file labeled 'Revised Manuscript with Track Changes'.An unmarked version of your revised paper without tracked changes. You should upload this as a separate file labeled 'Manuscript'.If applicable, we recommend that you deposit your laboratory protocols in protocols.io to enhance the reproducibility of your results. Protocols.io assigns your protocol its own identifier (DOI) so that it can be cited independently in the future. For instructions see: https://journals.plos.org/plosone/s/submission-guidelines#loc-laboratory-protocols. Additionally, PLOS ONE offers an option for publishing peer-reviewed Lab Protocol articles, which describe protocols hosted on protocols.io. Read more information on sharing protocols at https://plos.org/protocols?utm_medium=editorial-email&utm_source=authorletters&utm_campaign=protocols.

We look forward to receiving your revised manuscript.

Kind regards,

Fragkiskos Papadopoulos, Ph.D.

Academic Editor

PLOS ONE

Journal Requirements:

2. Please note that your Data Availability Statement is currently missing the repository name. If your manuscript is accepted for publication, you will be asked to provide these details on a very short timeline. We therefore suggest that you provide this information now, though we will not hold up the peer review process if you are unable.

**Additional Editor Comments:**

All three Reviewers have provided constructive comments for improving your manuscript. Please consider and address them as you see fit.

Reviewers' comments:

Reviewer's Responses to Questions

**Comments to the Author**

1. Is the manuscript technically sound, and do the data support the conclusions?

Reviewer #1: Yes

Reviewer #2: Partly

Reviewer #3: Yes

2. Has the statistical analysis been performed appropriately and rigorously? 

Reviewer #1: Yes

Reviewer #2: Yes

Reviewer #3: Yes

3. Have the authors made all data underlying the findings in their manuscript fully available?

Reviewer #1: Yes

Reviewer #2: Yes

Reviewer #3: Yes

4. Is the manuscript presented in an intelligible fashion and written in standard English?

Reviewer #1: Yes

Reviewer #2: Yes

Reviewer #3: Yes

5. Review Comments to the Author

Reviewer #1: This is a review of manuscript PONE-D-24-08069 titled “The importance of edge constraints in backbones of bipartite projections”. The authors show the importance of inducing the edge constraints in extracting the backbone from the bipartite network. With the empirical dataset of children’s play groups, they outline that backbone methods without any edge restrictions can lead to inaccurate representation of the system. Overall, the paper is well-written and, in my opinion, suitable for publication in PLOS ONE.

Here are some comments and questions that arose while I read the paper. The authors might want to address these if they feel it would enhance the manuscript---all up to them.

- Authors formulate the research questions as “To what extent do backbones extracted using a model that imposes edge constraints differ from backbones extracted using a model that ignores constrained edges?”. However, the comparison between the backbone in terms of topological features is missing. How do degree distribution, clustering coefficient, and other topological properties change when using SDSM-EC?

- The authors focus on one relatively small network (comprising 53 nodes). One could ask how important it is to impose the edge constrains in larger bipartite networks.

- For ease of reading, a subsection about the details of SDSM-EC could be added, making the paper self-contained. More information on the algorithm could help the paper understand more clearly.

- How does the backbone extracted using SDSM-EC differ from the one that first projects the bipartite network into the unipartite weighted network and applies the filtering, for instance, the disparity filter? In the latter case, the edge constraints are imposed by definition.

Reviewer #2: This paper underscores the necessity of imposing constraints on bipartite edges for accurate backbone extraction in empirical datasets. Utilizing a generic approach, it advocates for the adoption of models such as the Stochastic Degree Sequence Model with Edge Constraints (SDSM-EC) to enhance analytical precision.

The authors' conclusions are based solely on a single, small dataset, limiting the generalizability of their findings. To fortify the robustness of their claims, it is recommended that they incorporate additional datasets into their analysis.

Upon examining Table 3, age emerges as the most significant factor, supported by the authors' text, which highlights the natural grouping of children based on age. This observation is further corroborated by the confusion matrix, where false positives are notably lowest under a single constraint. However, further elaboration on this point from the authors is anticipated. Additionally, the necessity of considering additional datasets to strengthen their claims is emphasized. It would also be beneficial to report the confusion matrix in percentages or visualize it as a heatmap for improved readability.

Regarding a typographical error, "For example, an person cannot ..." should be corrected to "For example, a person cannot ..."

The caption for Figure 1 appears out of place. It would be helpful to clarify the representation of different shapes (e.g., squares, circles) in Figure 1.

Reviewer #3: In this article the authors provide a case study of the use of the Stochastic Degree Sequence Model with Edge Constraints (SDSM-EC) algorithm they have published previously, focusing on the improvement of backbone extraction of a bipartite network projection when an increasing number of edge constraints are present. They show that the quality of the backbone extraction of the projected bipartite network linearly scales with the number of constraints taken into account. I think the article gives a useful insight to complement previous work by the authors, and can be useful for researchers working with bipartite networks who need a way to include "forbidden" or "necessary" edges in the distribution probability. I think I have only a major comment, otherwise mostly minor ones.

My main reservation is that what the authors call "ground truth" is their most constrained model. So what they are looking at is how the results of their algorithm deviate from this most constrained case when a certain number of constraints are withdrawn / relaxed. However this ideal case is not compared to e.g. self-report data from children of who indeed they interact most with (which one could more easily associate to a "ground truth"), or compared with some accuracy with respect to "forbidden" interactions between children being in the backbone or not (i.e. building a unipartite scaffold that would just represent the constraints and checking how many times you cover the forbidden constraints). As such, I would expect this kind of analysis could have been better fulfilled using synthetic data, varying some proportion of 11 and 10 in the matrix (i.e. different constraints), and looking at the change in accuracy. For now, the paper is more of a "case study" and it is not clear how it generalizes and how it depends on the specifics of this particular dataset - which does not mean it is not useful to the community. Maybe that could be signalled in the title, adding that this is a case study using a children play dataset.

Minor comments:

p1 l12 an person -> a person

p2 l27: "We compare a backbone extracted using the SDSM-EC, and therefore yields a more accurate backbone"  yielding?

l51 : P = BB′ -> transpose is usually annotated with B^T, or B' should be defined

l58: the authors talk about the fact that projections are "noisy", but it seems they want to refer to the fact that the high density of links (i.e. having most Pij>0) limits the use of traditional network metrics that are more suited for sparse networks. It can be a bit confusing because it could give the impression that these edges are "false positives", when they are just an informational overload. It would be good if the authors explain a bit more that the noise here is in the sense of "outlier detection" (i.e. keeping only large values of edge weights) and that it requires a background model (i.e. what are the expected values at random).

Finally the code is very clean and useful. Just I want to mention that there is the line:

alpha <- .13 #Chosen based on https://doi.org/10.1038/s41598-021-03238-3

which is great but this should be in the Methods section, especially since the default in the library (and in general for statistical significance) is 0.05. Maybe a comment on this choice would be useful also (how would you results change with p=0.05 as is done often?).

6. PLOS authors have the option to publish the peer review history of their article (what does this mean?). If published, this will include your full peer review and any attached files.

Reviewer #1: No

Reviewer #2: No

Reviewer #3: No

---

## [Editor Report · Decision Letter 1]

17 Apr 2024

Illustrating the importance of edge constraints in backbones of bipartite projections

PONE-D-24-08069R1

Dear Dr. Neal,

We’re pleased to inform you that your manuscript has been judged scientifically suitable for publication and will be formally accepted for publication once it meets all outstanding technical requirements.

Kind regards,

Fragkiskos Papadopoulos, Ph.D.

Academic Editor

PLOS ONE
---

## [Editor Report · Acceptance letter]

29 Apr 2024

PONE-D-24-08069R1 

PLOS ONE

Dear Dr. Neal, 

I'm pleased to inform you that your manuscript has been deemed suitable for publication in PLOS ONE. Congratulations! Your manuscript is now being handed over to our production team.

Kind regards, 

on behalf of

Prof Fragkiskos Papadopoulos 

Academic Editor

PLOS ONE